# Performance Test and Microstructure of Modified PVC Aggregate-Hybrid Fiber Reinforced Engineering Cementitious Composite (ECC)

**DOI:** 10.3390/ma14081856

**Published:** 2021-04-08

**Authors:** Shi Hu, Haibing Cai, Rongbao Hong, Mengkai Li, Fangxing Yao

**Affiliations:** 1School of Civil Engineering and Architecture, Anhui University of Science and Technology, No. 168 Taifeng Road, Huainan 232001, China; aust_shihu@163.com (S.H.); cherishrb2020@163.com (R.H.); mengkaili2020@163.com (M.L.); m17775208836_1@163.com (F.Y.); 2Engineering Research Center of Underground Mine Construction, Ministry of Education, Anhui University of Science and Technology, No. 168 Taifeng Road, Huainan 232001, China

**Keywords:** engineered cementitious composites (ECC), hybrid fibers, recycled polyvinyl chloride (PVC) aggregate, impact loading, microstructure

## Abstract

This study aims to solve the problems of the high cost, heavy pollution and poor performance of traditional engineered cementitious composites (ECC) by adding modified Polyvinyl chloride (PVC) aggregate, Polypropylene (PP)–Polyvinyl alcohol (PVA) hybrid fiber and large amount of fly ash. The PVC aggregate is modified by pre-coating silica fume with a PP fiber volume content of 0.5%, PVA fiber volume contents of 1%, 1.5%, and 2%, PVC aggregate contents of 10%, 20%, and 30%, and fly ash volume content of 69%. Different properties and microstructures were studied by carrying out cube compression tests, splitting tensile tests, water absorption tests, drop hammer impact tests, scanning electron microscopy and nuclear magnetic resonance tests. According to the test results, under the same content of PVC aggregate, the use of modified PVC aggregate can, not only effectively avoid the decrease in strength and increase of water absorption, but also improve brittleness and impact failure energy. Regardless of the kind and content of fiber, the compressive strength and brittleness will decrease, while the splitting tensile strength, water absorption, and impact failure energy will increase. After adding 0.5% PP and 1.5% PVA fiber, the performance is ordinary and a negative mixing effect occurs. As more modified PVC aggregate is added, the strength of the ECC concrete with PP–PVA hybrid fiber and modified PVC aggregate added slowly decrease, while the water absorption and impact failure energy increase. Based on a comprehensive analysis of the test data, the reinforcement method of adding 1.5% PVA-0.5% PP hybrid fiber-30% modified PVC aggregate is superior to adding 1.5% PVA fiber, but slightly inferior to adding 2% PVA fiber. This study argues that the reinforcement method is of great significance for the promotion and application of ECC.

## 1. Introduction

As the most widely used building material, concrete has the advantages of abundant raw material sources, simple process, low production cost, fire resistance, high strength, good adaptability and convenient application. However, it also has some disadvantages, such as a large dead weight, brittle failure, large internal pores, and poor impact resistance [1]. In order to solve these problems, scholars have added different reinforcing fibers into concrete base materials. Marcalikova et al. [2] added different types and different amounts of steel fibers to prepare fiber reinforced concrete (FRC) and found that hooked fibers have better performance than straight fibers. As more fibers are added, the tensile strength and fracture energy increase. According to Brandt et al. [3], although different fibers were used, the fracture mechanic characteristics are not essentially different from those of ordinary concrete. The problems of brittle failure and cracking of concrete cannot be solved. They pointed out that ultra-high-performance fiber concrete is the future direction. Basalo et al. [4] pointed out that fire-resistant fiber reinforced cement-based materials are better than concrete with fire-resistant fibers that have been directly added. Pujadas et al. [5] studied the creep effect of adding plastic fiber. By comparing different creep characteristics of Polyethylene (PE) fiber and steel fiber, it was found that, after adding plastic fiber, the creep coefficient should be strictly controlled and kept at a low level to ensure structural safety. The fracture mechanic characteristics of prepared fiber reinforced concrete (FRC) are not essentially different from that of ordinary concrete. The problems of traditional concrete have not been solved [2,3,4,5]. In 1990, Professor Victor Li of the University of Michigan developed an engineered cementitious composite (ECC) with high ductility through a micromechanic design. There is no coarse aggregate in ECC, and ultra-fine sand with particle sizes of 100–200 mesh is used. Short and fine fibers with the volume ratio of 2%, fly ash, and water reducing agent are added [6]. The ultimate tensile strain of ECC is as high as 3%. Moreover, it has the advantages of good durability, impact resistance, and energy absorption [6,7,8]. In recent years, Li et al. [8] started to prepare ECC materials that meet different engineering requirements by adjusting the components and mix proportions, such as self-healing ECC, ultra-light ECC, and early-strength ECC. Scholars found that different fibers, such as Polyvinyl alcohol (PVA), Polyethylene (PE), Polypropylene (PP) and Polyethylene terephthalate (PET) can be added to ECC. Li et al. [9] found that the ultimate strain of 2% PVA fiber should be greater than 4%, the ultimate tensile strength is 4.5 MPa, and the crack width is controlled at 0.1 mm. 

Based on cost analysis and various performance metrics, it was found that PVA fiber produced by Kuraray Company of Japan (volume ratio is less than 2%) is the best ECC material combination [6,7,8,9], but the price is as high as 250 yuan/kg. Only by reducing the cost of PVA–ECC can it be widely used in China. Numerous studies have been carried out to effectively decrease the cost while maintaining or even improving the performance. Study methods are classified into three categories. First, some cheap fibers and PVA fibers are mixed and added to ECC, including PVA-steel fibers [10,11], PVA fibers made in foreign countries and China [12], PP-PVA fibers [13,14], etc. This method has been proved to have good performance [10,11,12,13,14]. Lin et al. [14] used PP and domestically made PVA fiber to reduce the cost and they studied the static and dynamic mechanical properties under different PP fiber contents while maintaining 1% PVA fiber unchanged. However, they did not study the if the content of PVA fiber is changed when the content of PP fiber is unchanged. The performance of 1.5% PP-1% PVA fiber is poor, especially the number of impact resistance is only 52. Second, the cost of cementitious materials in ECC decreases by reducing the amount of cement and using some environmentally friendly cementitious materials, such as fly ash. This method has been widely used [11,15,16,17]. Wang et al. [11] improved the strength of cementitious material by adding calcium sulfate, and the content of fly ash accounted for 60% of the cementitious material. The test strength met the design requirements. Zhang et al. [17] used fly ash that accounted for 80% of the cementitious material. Although the ductility was good at a high temperature, the strength was poor. Therefore, the amount of fly ash needs to be appropriate. The third category is to reduce the cost of aggregate in ECC, such as replacing quartz sand with ordinary river sand. However, the internal pores of ECC become large, and the performance poor [18,19]. Guan et al. [18] replaced quartz sand in traditional ECC with river sand, which decreased the cost by 10%, while maintaining a good tensile property. However, the internal pores became larger, and the strength obviously decreased. In recent years, it has been found that replacing ECC or natural fine aggregate in ordinary concrete with proper amounts of modified waste rubber or plastic aggregate can, not only prevent environmental pollution caused by rubber and plastic, but also significantly improve the ductility, impact resistance and energy absorption capacity of ECC [20,21,22]. Alaloul et al. [20] replaced quartz sand in traditional ECC with rubber particles and river sand, which decreased the strength and the elastic modulus. This method is suitable for non-load-bearing structures. Plastic is more difficult to decompose than rubber, and this causes serious pollution. Therefore, it is more meaningful to study replacing natural aggregates with PVC aggregates [22]. Although the three methods of reducing PVA–ECC cost noted above have achieved remarkable results, studies of effective combinations of the three methods are scarce. The basic mechanical properties, durability and impact resistance of the new ECC material are still unknown. It is worth studying whether adding modified PVC aggregate can reduce the negative impact of a hybrid fiber on PVA–ECC material.

In this study, the low cost, environmental friendliness, and good performance of PVA–ECC were realized by adding a large amount of fly ash, modified PVC aggregate, and PP–PVA hybrid fiber. The properties and microstructure of modified PVC aggregate–hybrid fiber reinforced ECC were studied by carrying out cube compression tests, splitting tensile tests, water absorption tests, drop hammer impact tests, scanning electron microscopy and nuclear magnetic resonance tests. Moreover, whether adding modified PVC aggregate can decrease the negative impact of hybrid fiber on PVA–ECC material was explored, expecting to provide a reference for mechanism studies and engineering applications of modified PVC-aggregate–hybrid-fiber-reinforced ECC.

## 2. Materials and Methods

### 2.1. Materials

The cement used was P.042.5 Portland cement produced by the Anhui Huainan Conch Cement Plant (Huainan, China). The fly ash was a first-class fly ash manufactured by the Henan Borun Material Company (Zhengzhou, China), and the specification was 3000 mesh. The fine aggregate was refined quartz sand with a size of 100–200 meshes, and the density was 1.79 g/cm^3^. The water was tap water from the laboratory. The water reducing agent was polycarboxylate superplasticizer produced by the Chenqi Chemical Company (Shanghai, China). The silica fume was high-activity silica fume produced by the Elkem Company of Norway (Shanghai, China). The PVC aggregate was recycled PVC particles with particle sizes of about 150 meshes and a density of 1.23g/cm^3^. The modified PVC aggregate was recycled PVC particles pre-coated with silica fume. In the pre-coating treatment, silica fume, water, and PVC particles were stirred in a portable mixer at low speed. The mass of the silica fume and water were 30% and 20% that of recycled PVC particles, respectively. The materials were air dried for 24 h, and then wrapped in plastic wrap and stored for 14 days [23,24]. The modification process is shown in Figure 1a. Polypropylene (PP) fiber was bundle monofilament polypropylene fiber produced by the Beijing Zhongjinkai Company (Shanghai, China), as shown in Figure 1b. The polyvinyl alcohol (PVA) fiber was RECS15/12 mm fiber, produced by the Kuraray Company (Okayama, Japan), as shown in Figure 1c. The physical performance indexes are shown in Table 1.

### 2.2. Specimen Preparation

At present, there is no specification on the mix proportion design of ECC materials. Based on domestic and foreign mix proportions and study results [6,7,8,9,10,11,12,13,14], this study designed the mix proportion shown in Table 2. PVC aggregate was added to ECC by replacing quartz sand with an equal volume, and different fibers were added to ECC by an external mixing method. Three groups of cementitious composites with 11 different mix ratios were designed.

Groups A, B and C were designed to study the modification effects of PVC aggregate, the fiber hybrid effect, and the effect of different modified PVC aggregate contents on B-5 (1.5% PVA–0.5% PP–ECC) (shown in Table 2).

The mixing amounts of cement, fly ash, water, and water reducing agent were 395 kg/m^3^, 869 kg/m^3^, 313 kg/m^3^ and 5.1 kg/m^3^, which are not repeated in Table 2. For the 11 kinds of cementitious composite materials with different mix ratios mentioned above, 9 cubic specimens with a size of 100 mm × 100 mm × 100 mm, and 3 cylindrical specimens with a size of 150 mm × 65 mm were prepared, with three specimens as a group. The arithmetic mean of the measured values of the three specimens was the final test result of a group.

The specimens were prepared as follows: firstly, quartz sand, PVC aggregate, cement, and fly ash were poured into a blender and dried for 4 min. Water and water reducing agent were added for wet mixing for 5 min. When the slurry flowed uniformly, fibers were slowly added and stirred for about 10 min until uniform a dispersion was achieved [6,7,8,9,10,11,12,13,14,15,16,17]. The mixture was poured into a mold, vibrated, and cured at room temperature for 36 h. Finally, the mixture was cured in standard curing room for 28 days.

### 2.3. Test Device and Methods

#### 2.3.1. Compressive Strength and Splitting Tensile Strength Test

The compressive strength of the cube specimens was tested by using a Suns WAW-1000 universal testing machine (Shanghai, China) at a loading speed of 3 mm/min. Cushion blocks were put above and below the specimen, and the splitting tensile strength of the cube specimen was tested at a loading speed of 1 mm/min.

#### 2.3.2. Water Absorption Test

Before the test, the cube specimen was put in an electric blast drying oven for more than 2 days, and the temperature was kept at 105 ± 5 °C. The specimen was weighed after cooling to room temperature. When the mass change between two consecutive days was smaller than 0.2% of the larger value, it was the final mass of the specimen. Four PVC pipes with a diameter of 20 mm were put in a sink to cushion the concrete specimen. Water was added to 30 mm higher than the top surface of the specimen. After keeping for more than 3 days, when the mass change between two consecutive days was smaller than 0.2% of the larger value, the specimen was taken out of water. The moisture on the surface was wiped with a wet cloth, and the saturated specimen was immediately weighed with an electronic balance [25]. The final water absorption formula is as follows:(1)$$W_{a}=\frac{m_{s}-m_{d}}{m_{d}}\times 100\%$$
where *W_a_* is the mass water absorption rate (%) of the specimen, correct to 0.1, *m_d_* is the mass (g) of the dried specimen. *m_s_* is the mass (g) of the saturated specimen.

#### 2.3.3. Drop Hammer Impact Test

The drop hammer impact test device is shown in Figure 2a. Before the test, the cylinder specimen was placed in a lower rigid baffle and it was fixed around the specimen with the paper baffle. A force transmission steel ball with a diameter of 63.5 mm was put into the positioning ring and an infrared induction counter was started. A steel wire rope was used to pull the cylindrical steel hammer with a diameter of 70 mm, height of 126 mm, and mass of 4.5 kg to 457 mm, as well as the impact of the force transmission steel ball. When an initial crack appeared, as shown in Figure 2b, the paper baffle was removed. When the specimen contacted any three of the four rigid baffles after cracking, it was regarded as completely damaged, and the number displayed on the counter was the number of impact times (*N*) during damage. To avoid the influence of air resistance and friction between the impact steel hammer and the sleeve, the optimization formula for calculating the impact failure energy (*E*) was obtained by combining the kinetic energy theorem [25,26].
(2)$$\begin{array}{c} E=N\frac{mv^{2}}{2}\\ where,\;v=\sqrt{2(0.9g)h} \end{array}$$
where *E* is the impact failure energy (J); *N* is the number of impact times during damage; *m* is 4.5 kg; *g* is 9.81 m/s^2^; *h* is 457 mm. At a coefficient of 0.9, the influence of air resistance and the friction between the impact steel hammer and sleeve in the test can be avoided.

#### 2.3.4. Scanning Electron Microscopy and Nuclear Magnetic Resonance Test

In the scanning electron microscopy (SEM) test, a scanning electron beam was used to excite physical signals of the sample surface and modulate the imaging, so that the microscopic three-dimensional morphology of the sample can be obtained. The concrete was cut in the core area of the compressive specimen into 6 mm × 6 mm × 4 mm slices. After drying the slices, the microstructure was studied using a Supra55 scanning electron microscope produced by Zeiss (Jena, Germany). In the nuclear magnetic resonance test, the specimen with a size of 2 mm × 2 mm × 2 mm was treated with vacuum saturation water and scanned using a MicroMR analyzer produced by the Niumag Company (Suzhou, China), and the internal pore structure information of the specimen was obtained. The water molecules in the magnetic field and the gradient field showed nuclear magnetic resonance phenomenon. The energy change signal of water molecules during resonance was converted into a T_2_ relaxation value, which was only related to the internal pore structure of the specimen. Therefore, the actual pore structure of the specimen could be obtained according to the change in the relaxation value [27].

## 3. Results and Discussion

### 3.1. Compressive Strength, Splitting Tensile Strength, Failure Process

The 28-day compressive strength, splitting tensile strength, elastic modulus values, and brittleness index value are shown in Table 3. The calculation formula of brittleness index B is as follows [28]:(3)$$B=\sigma_{c}/\sigma_{t}$$
where *σ_c_* is the compressive strength, *σ_t_* is the tensile strength, and the greater the *B* value the stronger the brittleness.

The load–displacement curves during uniaxial compression are shown in Figure 3, and the end point of the curve is the position where the post peak stress decreased to about 60% of the peak stress. It was found that the loading and unloading stages of the ECC matrix without fiber are very steep, especially in the unloading stage, which is close to a straight line because there is no fiber and coarse aggregate inside, only cementitious material, and quartz sand, similar to cement paste, which is thus very easy to crack. Before and after the addition of modified PVC aggregate, the slope of the loading curve decreased, and the downward trend was only slightly slower in the unloading stage; the addition of 0.5% PP fiber had little effect on the curve change, and the peak strain was higher than that of the ECC matrix. There was no obvious change in the unloading stage. With the increase in PVA fiber content, the slope of the loading curve obviously decreased and the peak strain increased. In group C, as the modified PVC aggregate increased, the slope of the loading curve decreased; the gentler decrease in the unloading stage was found along with a lowered peak strain.

In group A, the compressive strength and the splitting tensile strength showed similar trends, and the compressive strength and splitting tensile strength of the ECC matrix were 51.14 MPa and 1.84 MPa, respectively. Adding PVC aggregate (volume content was 30%) before and after modification decreased the strength of the ECC matrix. Adding unmodified PVC aggregate decreased the compressive strength and the splitting tensile strength of the ECC matrix by 19.01% and 17.53%, respectively; however, adding modified PVC aggregate only decreased the strength by 14% and 6.92%. Therefore, replacing PVC aggregate after special modification treatment can effectively delay the decrease in strength of the ECC matrix caused by directly adding PVC aggregate. According to the microstructure diagram in Figure 4, because of the side wall effect, the bonding effect in the interface transition zone between the cementitious material slurry and the PVC aggregate is poor, loose, and porous [29], but the silica fume on the surface of the modified PVC aggregate improved the bonding effect. However, the silica fume on the surface was uneven, and the silica fume and cement slurry were not tightly bonded, leading to the existence of an interface transition zone [30].

In group B, the influence of adding different types and different amounts of fibers on the strength of the ECC matrix was studied. It was found that adding 0.5% PP fiber decreased the compressive strength by 9.34% and increased the splitting tensile strength by 16.81%. With the increase in PVA fiber content, the compressive strength obviously decreased. When the volume content was 2%, the strength decreased by 18.65%. However, the splitting tensile strength increased notably, the maximum increase rate was 100.5%. According to the microstructure diagram in Figure 5, adding fiber can influence the compactness of the ECC matrix and weaken the bonding performance between aggregate and cementitious material slurry, resulting in a decrease in compressive strength. PVA fiber has a better hydrophilicity and fiberization than PP fiber and can be better bonded with cement slurry, leading to a slow decrease in compressive strength. In the failure process, the sliding friction between fiber and aggregate and cementitious slurry dissipated more energy. Thousands of fibers overlap and cross in the concrete, and a random support system is formed. The splitting tensile failure was more concentrated on the surface, which was different from the compressive failure. Therefore, the strength obviously increased [6,31]. In addition, the compressive strength and splitting tensile strength of hybrid fiber group B-5 were 39.07 MPa and 3.82 MPa. This was because the performance of PP fiber was inferior to that of PVA fiber. When 0.5% PP fiber was added, the compressive strength decreased by 9.98% and the splitting tensile strength only increased by 1.89% compared with the experimental group that had 1.5% PVA fiber added.

The influence law of 0.5% PP fiber and 1.5% PVA fiber on the strength was determined according to hybrid effect coefficient α. The formula is as follows [32]:(4)$$\alpha =\frac{\beta_{c,pva+pp}}{\beta_{c,pva}\beta_{c,pp}}$$

Hybrid effect coefficient α > 1 indicates a positive hybrid effect, otherwise it is a negative hybrid effect.

*β_c,pva_* is the strength ratio of the ECC with added PVA fibers. *β_c,pp_* is the strength ratio of the ECC with added PP fibers. *β_c,pva__+pp_* is the strength ratio of the ECC with added PVA and PP fibers. The formula of strength ratio *β_c_* is as follows [32]:(5)$$\beta_{c}=\frac{f_{f\text{-}ecc}}{f_{nc\text{-}ecc}}$$
where *f_f-ecc_* is the strength value of the ECC with added fiber. *f_nc-ecc_* is the strength value of the ECC matrix concrete.

Therefore, the B-5 hybrid effect coefficient, based on compressive strength, was 0.993 < 1, which was a negative hybrid effect. The B-5 hybrid effect coefficient based on splitting tensile strength was 0.882 < 1, which was a negative hybrid effect. In addition, the compressive strength and the splitting tensile strength of B-5 were 39.07 MPa and 3.82 MPa, respectively. Although both compressive strength and splitting tensile strength showed negative hybrid effects, the negative value was small, the compressive strength was close to a positive hybrid effect, and the splitting tensile strength was increased by 80.42% compared with the reference group. In addition, considering the low cost of PP fiber, the hybrid mode (B-5) is advisable.

In group C, as more modified PVC aggregate (10–30%) was added, the compressive strength and the splitting tensile strength of the ECC with 30% added PVC aggregate (C-3) were 9.7% and 5.17% lower compared with the hybrid fiber group concrete (B-5), respectively. Compared with the matrix ECC, the compressive strength decreased by 31.01%, and the splitting tensile strength increased by 71.1%, which were 35.28 MPa and 3.62 MPa, respectively. Although the silica fume on the surface of the modified PVC aggregate caused the bonding effect with cementitious material slurry to improve, there were still tiny pores, as shown in Figure 4, indicating the coexistence of hybrid fiber and modified PVC aggregate.

Typical compression failure modes and sketches of different concrete groups are given in Figure 6. Red lines with different widths are used according to the crack width, and the spalling parts are marked with yellow lines. Through the test, the crack widths can be divided into four types: large crack (>1500 μm), medium crack (1000–1500 μm), small crack (200–1000 μm), and micro crack (<200 μm). There are three types of cracks: through type, straight type, and bending type; the length of a crack is the sum of the lengths of the same type of crack widths.

It was found that the compression failure form of the ECC matrix (A-0) was similar to the brittle failure, and there were many macroscopic fracture surfaces. Replacing 30% natural aggregate with PVC aggregate, before and after modification, can reduce the elastic modulus and improve the brittleness (A-1, A-2). However, the modified PVC aggregate had a more obvious effect. This is because the silica fume on the surface of the modified PVC aggregate improved the bonding effect with the cementitious material slurry. In addition, the total length of large cracks, medium cracks, small cracks, and micro cracks of the A-0 and A-2 specimens were about 15.4 cm and 12.1 cm, 22.3 cm and 22.7 cm, 7.2 cm and 18.7 cm, and 3.1 cm and 4.1 cm, respectively. The medium and large cracks of the A-2 specimen decreased, while the small and micro cracks increased, which also verified that the brittle fracture of the A-2 specimen had improved to some extent.

After adding 0.5% PP fiber, there were fewer macro-penetrating fracture planes on the surface of the ECC (B-1) specimen, the elastic modulus decreased and the brittleness was improved. As more PVA fiber was added (1–2%), the number of macro-penetrating fracture planes on the specimen surface and elastic modulus gradually decreased, indicating that the brittleness was effectively improved. After adding 2% PVA fiber, the failure mode of the ECC (B-4) was the most complete. There was no penetrating fracture on the surface, and the minimum elastic modulus and brittleness basically disappeared. The failure mode of the ECC doped with 1.5% PVA + 0.5% PP fiber (B-5) was similar to that of the specimen with 1.5% PVA fiber (B-3) added, but the difference in elastic modulus between them was only 60 MPa, which also verified a similar failure mode. However, there were a few failure cracks on the compression surface. Its brittleness improvement effect (integrity of specimen) was slightly better than that of B-3, but much worse than that of B-4; the same trend was seen for the elastic modulus. The performance of PP fiber was inferior to that of the PVA fiber. Under the same content addition, PVA fiber could improve the brittleness of the ECC and the integrity of the specimen. In addition, the total lengths of the large crack, medium crack, small crack, and micro crack of the B-3 and B-5 specimens were about 15.1 cm and 11.7 cm, 10.7 cm and 11.9 cm, 7.6 cm and 16.7 cm, and 6.2 cm and 14.9 cm, respectively. It was found that the length of the medium crack and large crack were almost the same, while the length of small crack and micro crack B-5 was obviously greater, which also verified that the failure modes of the two specimens were similar, but the brittleness of B-5 was lower. The total lengths of the large crack, medium crack, small crack and micro crack of B-4 specimen were about 10.6 cm, 10.1 cm, 27.2 cm and 18.7 cm, respectively, so the integrity of B-5 specimen was better than B-3 and slightly worse than B-4.

Macroscopic penetrating fracture surface and spalling of the C-3 specimen (adding 30% modified PVC aggregate and 1.5% PVA + 0.5% PP fiber) were worse than the B-4 specimen (2% PVA), and the elastic modulus was second only to B-4, the brittleness improvement ability was second only to it. In addition, the total lengths of the C-3 test piece, medium crack, small cracks, and micro cracks were about 11.2 cm, 11.6 cm, 15.8 cm, 17.4 cm, respectively, which also verified the integrity of the C-3 test parts and above B-5, but below B-4

By observing the failure surfaces of specimens, with and without fibers, micro-cracks were found, which is of great help to understand the process of crack generation and development. Related pictures are shown in Figure 7.

In Figure 7a, it can be seen that when the crack propagates to the interface between quartz sand and cement paste, there is no obstacle, and it develops smoothly in a straight line, and the adjacent cracks will be connected. In Figure 7b, it can be seen that when the crack propagates to PVA fiber, the sliding friction of the fiber consumes a lot of energy, accompanied by many bending micro cracks. Therefore, the use of PVA fiber can enhance the crack resistance ability and transform larger cracks into many micro cracks.

According to the surface micro cracks, compression failure pattern sketch, and load–displacement curve of the failure specimen, the fracture process can be divided into two categories: the first category is group A and specimen B-1, and the fracture process can be regarded as brittle failure. There is an interface transition zone between natural aggregate, PVC aggregate, and cement paste. In the loading stage, the elastic modulus decreases, and the internal cracks or micro cracks expand like a straight line. The pores are connected. After reaching peak stress, the unloading stage is approximately straight. Internal cracks or micro cracks continue expanding. The pores are connected, forming longitudinal penetration cracks. Using modified and unmodified PVC aggregate and adding 0.5% PP fiber can alleviate the brittle failure of the ECC matrix. The second category is the remaining specimens with added PVA fiber. The fracture process can be regarded as a plastic-like failure. In the process of PVA fiber failure, more energy is dissipated because of sliding friction with the aggregate and cementitious material slurry interface. In the loading stage, the elastic modulus decreases, and the internal cracks and micro cracks develop from the interface transition zone between the cement paste, natural aggregate, and PVC aggregate. The micro cracks develop at the interface between the PVA fiber and cement paste. After reaching the bearing limit of PVA fiber, the micro cracks are connected. At peak stress, the fibers could still bear a certain load due to the tension in the unloading stage. Finally, when the elastic modulus becomes stable in the unloading stage, the PVA fibers will be pulled out or broken, and the cracks are connected, forming narrow penetration cracks. With the increase of the fiber content, there are fewer penetration cracks, and plastic failure is increasingly obvious.

Typical splitting tensile failure modes and sketches of the different concrete groups are shown in Figure 8.

According to the splitting tensile failure mode sketch and the observations of the test process, the fracture process can be divided into two categories: the first category is group A and specimen B-1, which are first in the elastic stage. With the increase in the load, the intermediate stress increases continuously and micro cracks appeared. As the load increased continuously, the cracks gradually developed near the pads at both ends, accompanied by a splitting sound. Finally, the specimens were split into two pieces. After using modified and unmodified PVC aggregate and 0.5% PP fiber, the splitting sound became smaller, and the main crack was increasingly narrow.

The second category was the remaining specimens with added PVA fiber. At first, with the increase in load, cracks appeared on the upper and lower end faces of quite a few parts. As the load continuously increased, the number and width of cracks gradually increased, but there was no penetration crack. Finally, all cracks were connected, forming longitudinal penetration cracks. However, the crack width was small, and there were connected PVA fibers in the cracks. With the increase in the fiber content, there were fewer cracks on the surface, and the micro cracks become narrower.

The above test process could be verified by the surface crack of the specimen. The total lengths of the large crack, medium crack, small crack and micro crack of specimens A-0 and B-1 were about 13.1 cm and 11.9 cm, 5.2 cm and 0.3 cm, 2.9 cm and 0.1 cm, and 1.5 cm and 0.9 cm, respectively, which also verified that this group of specimens was a complete split and the specimen was disconnected. Specimens B-4 and C-3 did not have large cracks, and the total lengths of the medium cracks, small cracks and micro cracks were about 1.5 cm and 4.9 cm, 32.9 cm and 28.6 cm, and 18.7 cm and 15.7 cm, respectively. It was verified that the surface transfixing cracks of the specimens doped with PVA fibers were less, or even non-existent, while the number of small cracks and micro cracks increased and the specimens did not break.

### 3.2. Water Absorption Rate

Harmful ions usually penetrate concrete through water, decreasing durability. Hence, preventing the penetration of water is one of most important methods to improve the durability of concrete, which is characterized by water absorption [33]. The water absorption rate of ECC is shown in Figure 9. Since the internal pores of the ECC materials are small, most scholars study capillary water absorption. For example, Da Costa et al. [34] carried out capillary water absorption tests and found that adding PP fiber to ECC can increase pores and water absorption. Zhang et al. [35] carried out a capillary water absorption test of ECC under different water–binder ratios, and the mass water absorption was about 2.35%. Considering the influence of errors, this study believed that the water absorption range meeting the design requirements was 2.2–2.5%.

In group A, ECC matrix (A-0) has the lowest water absorption rate of 1.599%. No fiber was added and quartz sand was used. The bonding effect was good and there were few pores. The water absorption of ECC increased by 2.178% and 1.925%, respectively, after adding PVC aggregate (30% volume) before and after modification, which indicates that replacing PVC aggregate with a special modification treatment can effectively decrease the porosity of the ECC matrix caused by directly adding PVC aggregate. The bonding effects of both modified and unmodified PVC aggregate were poor because of the side wall effect in the interfacial transition zone between the PVC aggregate and cementitious material slurry. The PVC aggregate was loose and porous; however, the silica fume on the surface of the modified PVC aggregate improved the bonding effect with the cementitious material slurry. The silica fume on the surface of the modified PVC aggregate was not uniformly wrapped however, and the silica fume and cement slurry were not closely bonded. Therefore, the water absorption rate was slightly improved.

In group B, the influence of different types and added contents of fibers on the water absorption of ECC matrix was studied. Adding 0.5% PP fiber increased the water absorption of the ECC matrix by 9.38%. As more PVA fiber was added, the water absorption increased significantly. When the volume content of PVA fiber was 2%, the water absorption was increased by 29.89%. The water absorption was increased by 12.57% when the volume content was 1%. Compared with the increase of 9.38% caused by adding 0.5% PP fiber, PVA fiber had a better performance than PP fiber under the same volume content. No matter what kind of fiber was added, the compactness of the ECC matrix was influenced, and the bonding performance of aggregate and cementitious material slurry was weakened, thus increasing the water absorption. However, PVA fiber has a better hydrophilicity and fiber-forming ability compared to PP fiber. Therefore, the compactness decrease effect was poor, with few pores and a slow increase in water absorption. In order to study the influence of adding 0.5% PP fiber and 1.5% PVA fiber (B-5) on water absorption, the hybrid effect coefficient was calculated to be 0.987 < 1 according Equations (4) and (5), but the water absorption was 2.139%. The negative effects of the two fibers on water absorption were not superimposed or even slightly decreased after mixing. Therefore, the hybrid mode had a positive hybrid effect. Water absorption was negatively correlated with durability. Although the coefficient of hybrid effect was smaller than 1, a lower water absorption is suitable. As for durability, we believe that the hybrid mode was a positive hybrid effect. Although the water absorption of the concrete increased after composite modification, there were many factors influencing the effect of composite modification, such as fiber distribution, test error, temperature, and humidity. Therefore, based on the water absorption performance, the hybrid method (B-5) is advisable.

In group C, with the increase in content of added modified PVC aggregate (10–30%), the water absorption of ECC (C-3) with 30% added PVC aggregate reached 2.441%, which was 14.12% higher than that of hybrid fiber group B-5 and 52.66% higher than that of matrix ECC. Although the silica fume on the surface of the modified PVC aggregate improved the bonding effect between the modified PVC aggregate and the cementitious slurry, there were still micro pores because C–S–H gel is generated in the secondary hydration of silica fume and Ca(OH)_2_; the harmful pores in hardened cement paste were filled. The distribution of silica fume coated on the surface of PVC aggregate was uneven. As shown in Figure 4d, there are still tiny pores [36,37]. More pores exist in the hybrid fiber. Because water absorption is closely related to the internal pore structure of the specimen [27], the information of the internal pore structure of the specimen was studied using NMR experiments. The results are presented in Figure 10.

The abscissa of the T_2_ spectrum distribution curve is related to the pore radius, and the ordinate is related to the number of pores. The longer the transverse relaxation time (T_2_), the larger the pore radius. As the amplitude of the longitudinal signal increases, there are more pores under the corresponding pore radius.

There are three peaks in the T_2_ spectrum distribution curve, which are small, medium and large pores, respectively. The three peaks show the same change trends. By calculating the T_2_ spectrum area of the specimen, it was found that the T_2_ spectrum area of B-4 increased by 20.9% compared with that of A-0, and the actual water absorption increase was 29.8%. The T_2_ spectrum area of C-3 was 49.8% greater than that of A-0, and the actual water absorption increase was 52.7%. Adding fiber and PVC aggregate could enlarge the pore structure, and the PVA fiber mainly led to an increase of small pores. PVC aggregate led to the appearance of medium and large pores. In addition, the water absorption was closely related to the pore structure. The nuclear magnetic resonance experiment results were similar to the water absorption experiment results, which also proved that the experimental results are correct.

### 3.3. Impact Failure Energy

The impact failure energy of different ECCs are shown in Figure 11. In projects where impact loads often occurs, such as military protection, deep roadways, tunnel supports, etc., it is necessary to use concrete with strong impact resistance and energy absorption ability [38,39].

In group A, the impact failure energy of the ECC matrix (A-0) was only 54.58 J. No fiber was added, there was no natural coarse aggregate, and the brittleness was large. As shown in Figure 12a, the failure surface developed rapidly and penetrated thoroughly. The impact failure energy of ECC increased significantly after adding PVC aggregate (volume content 30%) before and after modification, which were 72.64 J and 108.96 J, respectively, and increased by 33.09% and 99.63%. Using PVC aggregate with special modification can effectively decrease the porosity of the ECC matrix caused by directly adding PVC aggregate so that the impact resistance of the ECC is improved. Recycled PVC plastic has good elasticity and low shock wave impedance. Under an impact load, the PVC particles deform, which causes an unloading effect on the shock wave, and the impact energy of a falling hammer is consumed. In addition, as the internal micro cracks of the concrete develop, a large deformation of the PVC particles can relieve the stress at the crack tip, avoid the stress concentration and delay crack propagation. In addition, the silica fume on the surface of PVC aggregate improved the bonding effect and impact resistance.

In group B, the impact failure energy of the ECC matrix with different types and amounts of added fibers was studied. The impact failure energy of PP fiber with a volume content of 0.5% was increased by 133.33% compared with ECC matrix, but the brittleness improvement effect was general. The failure mode was similar to that of the ECC matrix group, as shown in Figure 12b. As more PVA fibers were added, the impact failure energy increased significantly. When the volume content of the PVA fiber was 2%, the impact failure energy increased by 276 times. The impact failure energy increased by 58 times when the volume content was 1%. Compared with the 133.33% increase under 0.5% volume content of PP fiber, the performance of PVA fiber was much better than that of PP fiber. Fiber has a good impact resistance performance, and polypropylene fibers in the concrete were overlapped and implicated with each other, forming a disordered supporting system. Under the impact load, there were micro cracks in the concrete, the directions of which were random. In the process of micro crack propagation, the load was transferred to the fiber bridged at the cracks. However, PVA fiber has better hydrophilicity and fibrillation than PP fiber, and the effect of decreasing compactness was poor. There are fewer pores, and the impact failure energy increased significantly. The failure mode of the ECC with 2% added PVA fiber is shown in Figure 12c. The specimen had the best integrity, and the failure mode was similar to that in the reference. As shown in Figure 12d, the PVA fibers on the fracture surface were evenly dispersed and still maintained a good implicative effect. In addition, after recording the failure times, the specimen was smashed until the PVA fibers on the fracture surface were broken. However, the test block still had good integrity, as shown in Figure 12e, indicating the excellent performance of PVA fibers.

In order to study the influence law of adding 0.5% PP fiber and 1.5% PVA fiber (B-5) on the impact failure energy, the hybrid effect coefficient was calculated to be 0.601 < 1, according to Equations (4) and (5), which was a negative hybrid effect. The impact failure energy reached 9751.92 J. This was because the impact failure energy of the composite modified concrete was enhanced. However, the reinforcement effect of the composite modification was influenced by different factors, which was not simple superposition. Considering the low cost of PP fiber and the obvious increase in impact failure energy after mixing, the hybrid mixing method was beneficial for engineering applications of PVA fibers.

In group C, as more modified PVC aggregate was added (10–30%), the impact failure energy increased. Compared with the hybrid fiber group (B-5), the impact failure energy of ECC (C-3) with 30% added PVC aggregate was 13,747.12 J, increased by 40.97%, and was 251 times higher than that of matrix ECC. The failure mode is shown in Figure 12f, which is similar to that of B-4 (2% PVA), but there are few secondary cracks. This also corresponds to the actual impact damage energy value. The impact failure energy of ECC is enhanced after adding hybrid fiber and modified PVC aggregate. There were many factors influencing the composite reinforcement effect, which is not simple superposition. The failure impact energy of the concrete specimen was positively correlated with the number of damaged specimens [25,39].

According to the fracture morphology diagram of the impact of a falling hammer and the observations during the test, the fracture process under the impact of a falling hammer can be divided into three categories: the first category is the A-0 specimen. The initial cracks develop linearly and break into two large pieces. The second category is specimens A-1, A-2 and B-1. The initial cracks appear in three directions. If the impact continues, the circular pit on the surface of the specimen will become larger, and the crack width in the initial crack direction will become larger. The specimen will be divided into three pieces, which are connected. However, the connection strength will decrease under continuous impact. The third category is the remaining specimens with added PVA fiber. The initial cracks appeared in different directions. If the impact continues, the circular pits and depths on the surface of the specimens become large. There are a large number of micro cracks around the pits. With a large gap and several wide cracks good integrity is formed. In addition, with the increase of PVA fiber content, except for a big gap, the larger the fiber content, the wider the cracks.

## 4. Discussion

### 4.1. Size Effect

The size effect of concrete materials refers to the phenomenon that the strength, brittleness index and other properties of specimens are no longer constant but change with the geometric size of specimens. The size effect has the greatest influence on the strength. In order to solve the influence of the size effect, according to the Chinese standard, standard cube specimens with a side length of 150 mm were used to evaluate the strength grade of concrete. When non-standard specimens with side lengths of 200 mm and 100 mm are used, they are multiplied by size conversion factors of 1.05 and 0.95, respectively.

ECC is a new type of concrete, which is quite different from ordinary concrete in material composition, performance, and preparation method. In order to accurately judge the strength, bearing capacity, and durability of structural concrete and ensure that the test results have wide applicability and practical significance, we must study the size effect and determine the standard specimen size and conversion coefficient [40].

### 4.2. Shrinkage Performance

Drying shrinkage is the most common among all kinds of shrinkage deformations, and the deformation is large. The drying shrinkage of ECC is three times that of ordinary concrete under the same environmental conditions. In practice, ECC has a large shrinkage deformation, which causes cracks and affects the durability of the structure [41]. Although using hybrid fiber can reduce the drying shrinkage, adding PVC aggregate will increase the drying shrinkage. Therefore, it is necessary to consider the shrinkage deformation value under the mixing mode. Different components and proportions should be designed according to construction environments.

### 4.3. Creep and Rheological Properties

In order to design the stress and deformation of the ECC structure safely and economically for the whole working period, the creep properties of materials should be considered. It is of great significance to study creep and relevant calculation theories. Li et al. [42] found that when the water–binder ratio is 0.25 and the PVA fiber content is 2%, about 70% of creep is completed in 7 days, and the creep degree reaches 20 after 120 days, which meets engineering requirements. Since PP fiber influences the creep of concrete more obviously than PVA fiber, the creep properties under the mixing mode should be considered to avoid engineering accidents.

It is of great importance to study the rheological properties of fresh concrete for ECC engineering applications. Sahmaran et al. [43] found that when the water–binder ratio is 0.27, the sand–binder ratio is 0.36, the water reducing agent dosage is 4.2 kg/m^3^, and 2% PVA fiber is used, the mini-slump flow diameter is 188 mm. However, PVC aggregate can improve the rheological properties. Different materials lead to different rheological effects and the pouring and slump test diagram is shown in the Figure 13. The ECC slurry is viscous, with large fluidity, and the slump is 168 mm, which meets the requirements of sprayed concrete. So, the components and mixing proportions should be reasonably adjusted according to the rheological properties of different engineering backgrounds.

## 5. Conclusions

This study investigated the properties and microstructure of multi-reinforced ECC. The following conclusions could be obtained:The strength of the ECC matrix with added PVC aggregate before and after modification decreases but adding modified PVC aggregate can effectively delay the effect of directly adding PVC aggregate on strength reduction. The results demonstrate that the compressive strength of the ECC matrix decreases as 0.5% PP fiber is added, but the splitting tensile strength increases. With the increase of the PVA fiber content, the compressive strength decreases significantly, which is 18.65% when the volume content is 2%, but the splitting tensile strength is increased by 100.5%. The hybrid effect coefficients based on the compressive strength and the splitting tensile strength of 0.5% PP and 1.5% PVA fiber groups are 0.993 and 0.882, respectively. The compressive strength and the splitting tensile strength of the group added with hybrid fiber-modified PVC aggregate decreases slowly as more modified PVC aggregate is added, which are 35.28 MPa and 3.62 MPa, respectively.The compressive failure mode of the ECC matrix is similar to brittle failure. Adding PVC aggregate before and after modification can decrease the elastic modulus and effectively improve brittleness, but modified PVC aggregate has a more obvious effect. With the increase in the PVA fiber content, there are fewer macro penetration fracture planes on the specimen surface and the elastic modulus decreases, indicating that the brittleness has been effectively improved. As 2% PVA fiber is added, the failure form is the most complete, there is no penetration fracture plane, and the brittleness basically disappears. The failure mode of adding 1.5% PVA and 0.5% PP fibers is similar to that of adding 1.5% PVA fiber. However, there are few cracks on the compression surface, and the elastic modulus is a little bit lower, which indicates that its brittleness improvement effect is slightly better than that of 1.5% PVA fiber, but much worse than that of 2% PVA fiber. The macro penetration fracture plane and spalling of specimens with added hybrid fiber and 30% modified PVC aggregate are worse than that added with 2% PVA fiber, the elastic modulus and the brittleness improvement ability is second only to it.The water absorption of the ECC matrix with added modified and unmodified PVC aggregate increases. Adding modified PVC aggregate can effectively decrease the porosity caused by directly mixing PVC aggregate. Regardless of the kind of fiber and the content, the water absorption increases. As 1.5% PVA and 0.5% PP fibers are added, the water absorption is the largest, and the hybrid effect coefficient based on water absorption is 0.987. As more modified PVC aggregate is added, the water absorption of the ECC added with hybrid fiber and modified PVC aggregate increases, reaching 2.441%.ECC matrix has a large brittleness, and the impact failure energy with the added modified and unmodified PVC aggregate obviously increases. The modified PVC aggregate has a more significant reinforcement effect. The impact failure energy of 0.5% PP fiber is increased by 1.33 times, the effect is general. However, PVA fiber increases the impact failure energy more obviously. The impact failure energy is increased nearly by 276 times when the mixing content of PVA fiber is 2%. The hybrid effect coefficient of 0.5% PP and 1.5% PVA fibers based on impact failure energy is 0.601. The impact failure energy of the ECC added with hybrid fiber and modified PVC aggregate increases as more modified PVC aggregate is added, reaching 13,747.12 J.

According to a comprehensive analysis of the experimental results above, although the treatment method of 0.5% PP–1.5% PVA hybrid fiber–30% modified PVC aggregate increased the water absorption of the ECC matrix, it was smaller than that of ordinary concrete. The compressive strength and the splitting tensile strength decreased, but they were still 35.28 MPa and 3.62 MPa, which can meet the requirements of practical engineering. The brittleness improvement effect was enhanced. Considering the low cost, environmental protection, and high performance, 1.5% PVA–0.5% PP hybrid fiber–30% modified PVC aggregate has great significance to the popularization and application of ECC.

## Figures and Tables

**Figure 1 materials-14-01856-f001:**
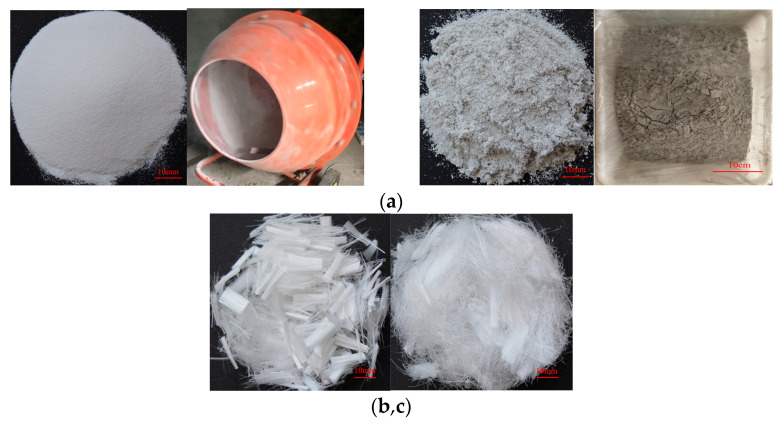
Materials: (**a**) Modified Polyvinyl chloride (PVC) aggregate treatment process; (**b**) Polypropylene (PP) fibers; (**c**) Polyvinyl alcohol (PVA) fibers.

**Figure 2 materials-14-01856-f002:**
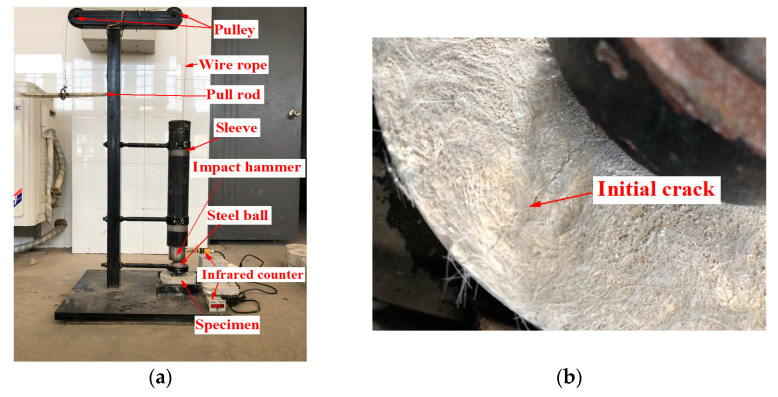
Impact test: (**a**) Self-made impact test device; (**b**) initial crack.

**Figure 3 materials-14-01856-f003:**
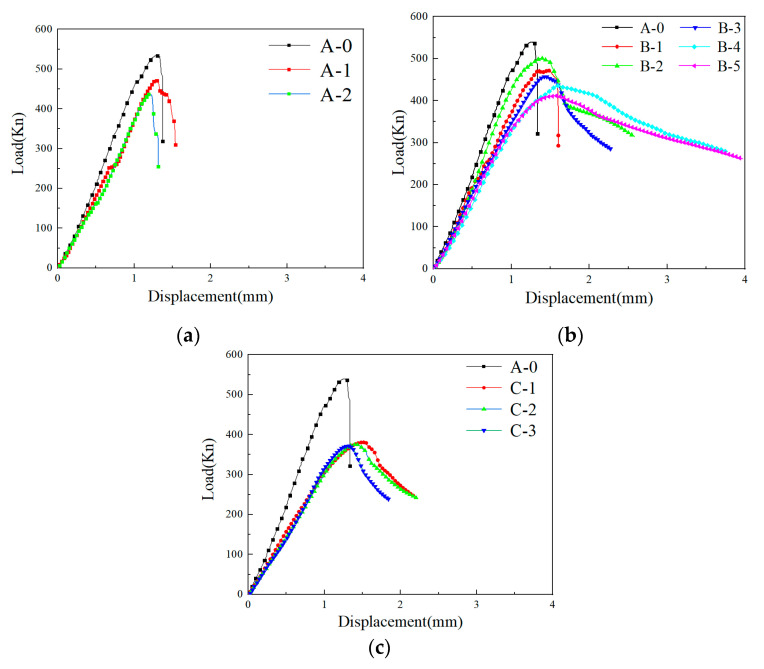
The load–displacement curve: (**a**) Group A; (**b**) group B; (**c**) group C.

**Figure 4 materials-14-01856-f004:**
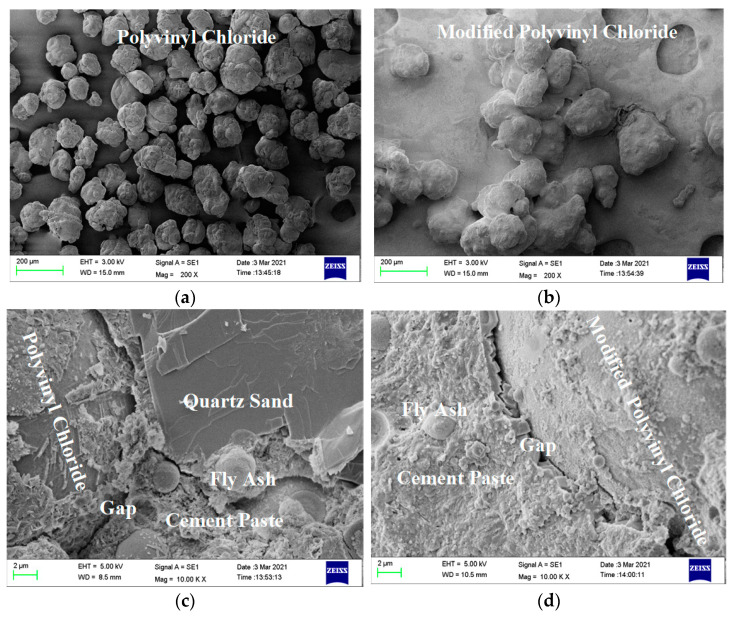
Microstructural diagram. (**a**) Unmodified PVC aggregate surface (200×); (**b**) modified PVC aggregate surface (200×); (**c**) gap in Specimen A-1 (10K×); (**d**) gap in Specimen A-2 (10K×).

**Figure 5 materials-14-01856-f005:**
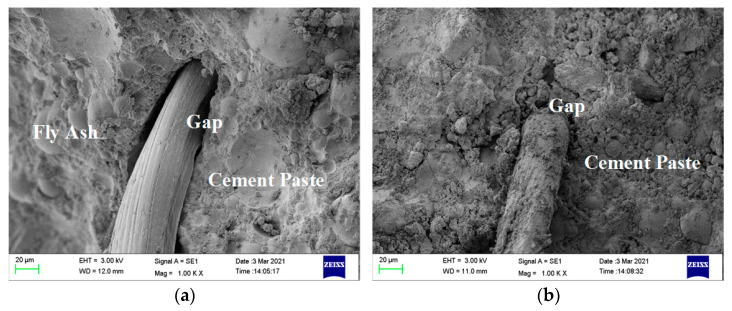
Microstructure diagram of fibers. (**a**) PP fiber (1K×); (**b**) PVA fiber (1K×).

**Figure 6 materials-14-01856-f006:**
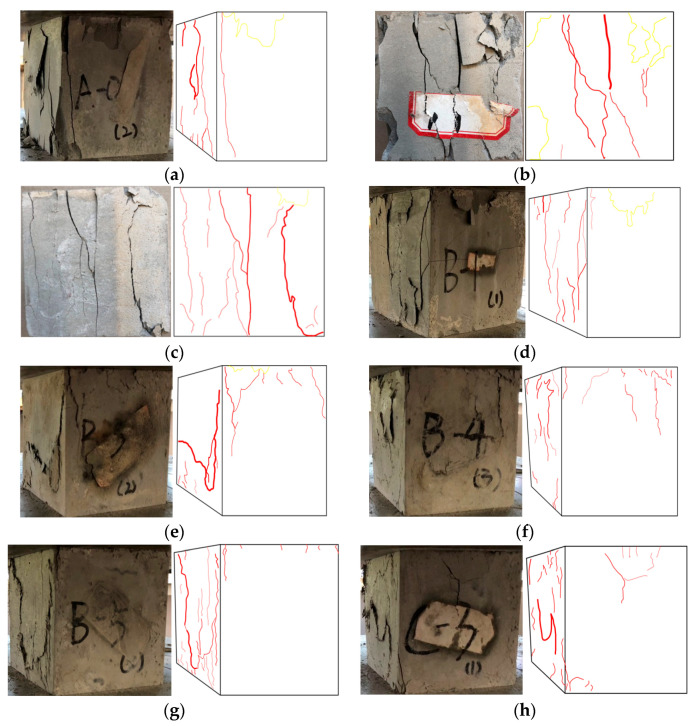
Typical compression failure modes and sketches of concrete. (**a**) A-0; (**b**) A-1; (**c**) A-2; (**d**) B-1; (**e**) B-3; (**f**) B-4; (**g**) B-5; (**h**) C-3.

**Figure 7 materials-14-01856-f007:**
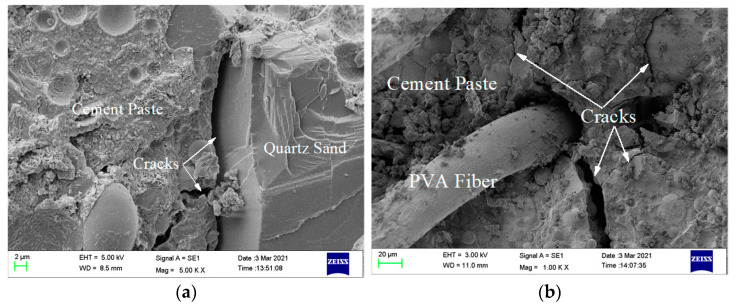
Microstructure diagram of cracks. (**a**) Without fibers (5K×); (**b**) PVA fiber (1K×).

**Figure 8 materials-14-01856-f008:**
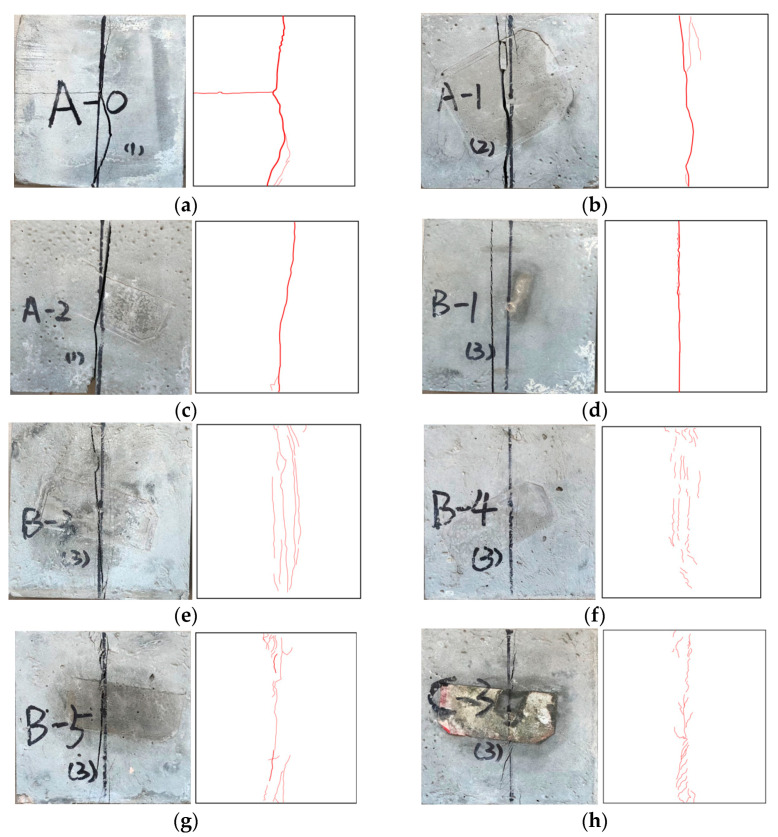
Typical splitting tensile failure modes and sketches of concrete. (**a**) A-0; (**b**) A-1; (**c**) A-2; (**d**) B-1; (**e**) B-3; (**f**) B-4; (**g**) B-5; (**h**) C-3.

**Figure 9 materials-14-01856-f009:**
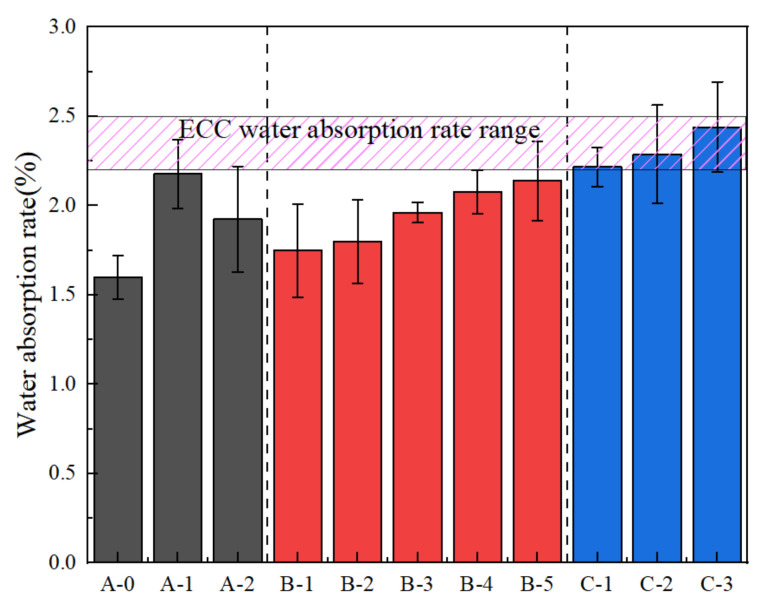
Water absorption rate.

**Figure 10 materials-14-01856-f010:**
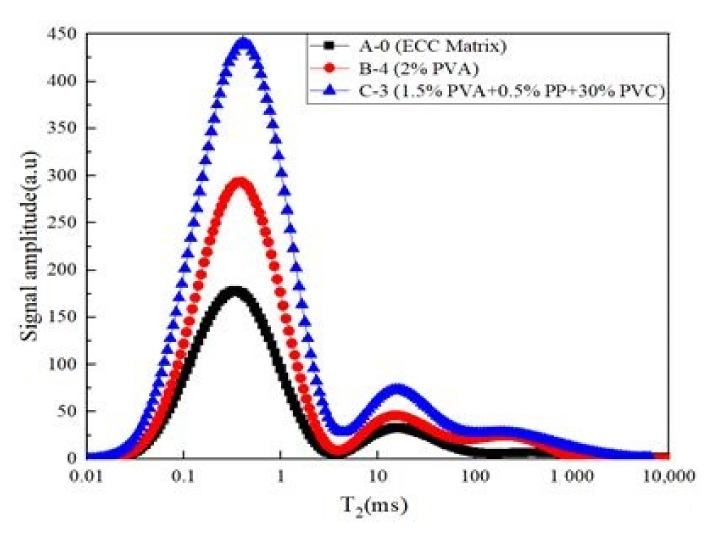
Distribution curve of nuclear magnetic resonance T_2_ spectrum.

**Figure 11 materials-14-01856-f011:**
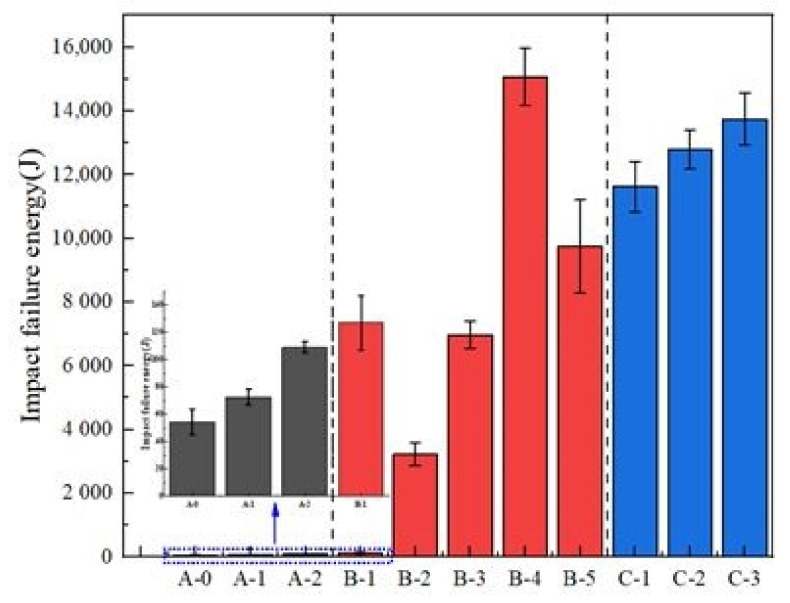
Impact failure energy.

**Figure 12 materials-14-01856-f012:**
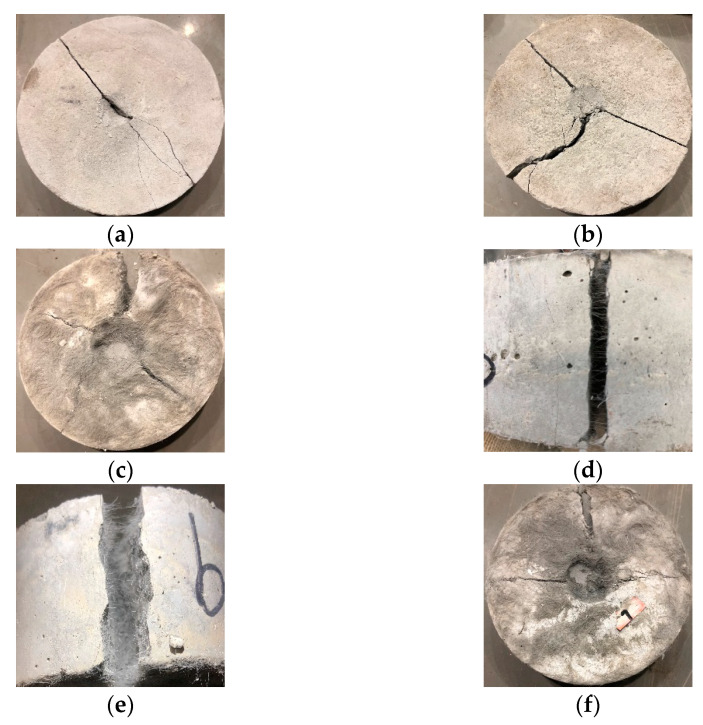
Typical drop hammer impact failure forms of concrete and rupture surface. (**a**) A-0; (**b**) B-1; (**c**) B-4; (**d**) PVA fibers on fracture surface; (**e**) fracture surface of broken PVA fibers; (**f**) C-3.

**Figure 13 materials-14-01856-f013:**
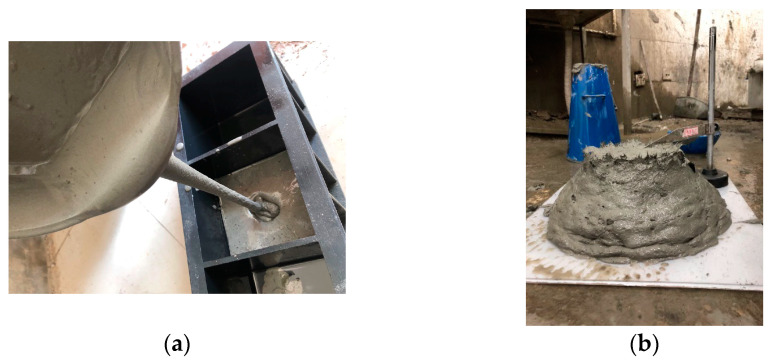
Engineered cementitious composite (ECC) pouring and slump test. (**a**) Pouring; (**b**) slump test.

**Table 1 materials-14-01856-t001:** Physical properties of fibers.

Species	Length	Diameter	Density	Elastic Modulus	Tensile Strength	Elongation at Break
PP Fiber	12 mm	18 μm	0.91 g/cm^3^	4.5 GPa	500 MPa	26.8%
PVA Fiber	12 mm	40 μm	1.3 g/cm^3^	42.8 GPa	1560 MPa	6.5%

**Table 2 materials-14-01856-t002:** Concrete mix proportions.

Specimen	A-0	A-1	A-2	B-1	B-2	B-3	B-4	B-5	C-1	C-2	C-3
PVC aggregate (kg/m^3^)	0	94.5	0	0	0	0	0	0	0	0	0
Modified PVC aggregate (kg/m^3^)	0	0	94.5	0	0	0	0	0	31.5	63	94.5
PVA fiber (kg/m^3^)	0	0	0	0	13	19.5	26	19.5	19.5	19.5	19.5
PP fiber (kg/m^3^)	0	0	0	4.55	0	0	0	4.55	4.55	4.55	4.55
Sand (kg/m^3^)	459	321.3	321.3	459	459	459	459	459	413.1	367.2	321.3

**Table 3 materials-14-01856-t003:** Compressive strength, splitting tensile strength, elastic modulus, brittleness index value and their standard deviation.

Specimen	A-0	A-1	A-2	B-1	B-2	B-3	B-4	B-5	C-1	C-2	C-3
Compressive strength (MPa)	51.14	41.42	43.98	46.36	47.51	43.4	41.6	39.07	36.16	35.95	35.28
Standard deviation	5.07	3.76	2.45	4.95	5.23	4.89	3.10	2.98	3.02	4.32	2.13
Splitting tensile strength (MPa)	1.84	1.52	1.71	2.47	3.4	3.75	4.25	3.82	3.73	3.72	3.67
Standard deviation	0.21	0.26	0.34	0.21	0.32	0.28	0.31	0.19	0.24	0.20	0.26
Elastic modulus (GPa)	4.06	3.51	3.46	3.58	3.13	3.06	2.91	3.03	3.02	2.97	2.95
Standard deviation	0.26	0.34	0.41	0.42	0.39	0.27	0.42	0.21	0.37	0.19	0.36
Brittleness index value	27.79	27.25	25.71	18.77	13.97	11.57	9.79	10.23	9.73	9.66	9.61

## Data Availability

The data presented in this study are available on request from corresponding author.

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
