# Peer review of "Performance Test and Microstructure of Modified PVC Aggregate-Hybrid Fiber Reinforced Engineering Cementitious Composite (ECC)"

_materials, 2021, doi:10.3390/ma14081856_

Round 1

Reviewer 1 Report

The current study investigates adding modified PVC aggregate and PP-PVA hybrid fiber in cementitious composite. The authors prepared three types of specimens to determine the PVC and fibre effects on mechanical properties such as tensile, impact, SEM, nuclear magnetic resonance and water absorption. The authors found that adding PVC aggregate reduced the strength of the matrix and increased its ductility and water absorption and impact resistance. Adding modified PCV showed somewhat further improved characteristics.

The abstract is good but needs to be shortened and be precise, I got a bit lost while reading what did the authors do in this study, make it simple, what did you do, what did you find and how did this benefit and affect the ECC performance. Therefore, please consider reviewing the abstract and highlight the novelty, major findings and conclusions.

Introduction is brief and needs more work, what is the research gap did you find from the previous researchers in your field? Mention it properly. It will improve the strength of the article.

Combine figures 1 and 2 into one figure and add scale bar for all images

Table 1 need a reference if not measured by the authors

Figure 3 add some arrows and text to explain to us what are we looking at in here

How many times each samples was repeated for each test? Figure 6 is missing error bars, did you just do each test once? This is not enough to confirm repeatability of your tests

Authors should add a table/list of nomenclatures and symbols at the end or start of the manuscript

Line 306 so what does this tell us about this sample? Please explain and support with references

Figure 10 what is the critical water absorption level for this material or the application it will be used for, draw red line at the top of the bar chart to show us which ones of these tested samples exceed the allowable limits

Figure 10 how about results from previous studies what did they find, is it similar to your work or different, please discuss and analyse your work and compare it with past studies and support with references

Line 351-352 “Therefore, the hybrid mode is positive hybrid effect.” I don’t understand this sentence please elaborate further

Line 354 “there are many factors influencing the effect of composite modification” such as? Don’t write incomplete sentence and leave it like this it makes the paper lack something important

Line 362 “there are still micro pores.” Why? Explain and support with references

The results are merely described and is limited to comparing the experimental observation. The authors are encouraged to include detailed discussion and critically discuss the observations from this investigation with existing literature.

Author Response

Please see the attachment,THANKS

Reviewer 2 Report

The originality and the scientific value of the subject research are good.

The research area is Performance test and microstructure of modified PVC aggregate-hybrid fibre-reinforced engineering cementitious composite.

The manuscript presents a series of experiments that are logical.

Some research information is already known.

For the experimental program, it would be more appropriate to use a three-point bending test to determine the load-displacement diagram and the fracture energy or residual tensile strength.

However, the selected variant of the split tensile test is possible.

The manuscript has the usual structure.

I recommend improving the introduction section. There is extensive research in the solved area (fibre-reinforced concrete and engineering cementitious composite), which needs to be better explained within the solved research. Among the interesting reviews and articles from recent and solved areas are:

Marcalikova, Z. et al. Determination of Mechanical Characteristics for Fiber-Reinforced Concrete with Straight and Hooked Fibers. Crystals 2020, 10, 545. 
Brandt, A.M. Fiber reinforced cement-based (FRC) composites after over 40 years of development in building and civil engineering. Compos. Struct. 2008, 86, 3–9.

and many others.

In the experimental program, it is missing to state the Modulus of elasticity.
It would be appropriate to arrange tests focused on durability and longevity.
For a comprehensive solution to the problem, it would also be appropriate to use numerical modelling of experiments.

The tests results should be presented in the context of current research. I recommend expanding the discussion and conclusion section. It is necessary to improve the presentation of new knowledge and the added value of the article for further research.

It is necessary to check the template and carefully prepare the manuscript. I also recommend checking English.

The description should be on the next page. (Table 2.)
Variables (for example - E,...........) should be in italics.  - applies to all manuscript.
The label is on the wrong side (Figure 5)
The text in Figure 5a is illegible - edit.

The tests results must present also (in tables/figures) the standard deviation, the number of samples and the CoV. - applies to all manuscript.

Figure 7. - state the size of the magnification in the caption of the image.

Figure 8. - state the size of the magnification in the caption of the image.

The label is on the wrong side (Figure 12).

The manuscript needs to be revised.

Author Response

Please see the attachment,THANKS.

Reviewer 3 Report

The manuscript represents a valuable contribution in terms of experimental results but clearly fail in terms of novelty and originality. 

  • The authors should better emphasis the research significance of the manuscript and their contribution in terms of novelty
  • The state of the art carried out in the introduction is short and poor, please improve it by including further references
  • The addition of plastic fibres may bring concrete an additional problem, creep. The authors should at least warn the reader about this fact in the manuscript (check and cite if necessary https://doi.org/10.1016/j.conbuildmat.2017.05.166 )
  • Studies have shown that an anisotropic orientation of fibres in the hardened state may occur as a result of the different stages and processes (fresh state, the concrete pouring, the geometry of the formwork, the type of vibration and the production method) that FRC undergoes from mixing to hardening, however the authors does not include any detail regarding this stages. Authors should include details of how concrete was mixed and poured, as well as fresh state properties of concrete in order to understand the results later presented.
  • Authors should include additional test to show the post-craking behaviour of the material. For practical reasons, the reference standards and codes recommend the use of flexural tests on a prismatic specimen either in the configuration of a three-point bending test or the four-point bending test. Despite being widely used these tests exhibit a high scatter of the results, often above 20%, which compromises its use as a control tool. Other standard methods have also been used for the material characterization, namely: round determinate panel tests, EFNARC panel test and wedge-splitting tensile test. The latter was recently modified by introducing a wedge-shaped cut and a notch (Double Edge Wedge Splitting or DEWS) aiming at reducing the scatter of the results. Nevertheless, its associated labour costs and complexity hinders the use of the DEWS test for systematic control and design purposes. However the most suitable for your research is de Double Punch Test (so called Barcelona Test) and particularly its alternative in cubic specimens called Multidirectional double punch test (MDPT) which you may find in ( https://doi.org/10.1016/j.conbuildmat.2014.02.023 )
  • Post-cracking behaviour of concrete in compression could also be an interesting output. The conclusions driven on this issue by this manuscript are well known, the authors should try to include conclusions beyond what has been already published
  • Please review English

Author Response

Please see the attachment,THANKS.

Round 2

Reviewer 1 Report

All questions answered 

Author Response

Point 1: All questions answered 

Response 1: Thank you for your comments.

Reviewer 2 Report

The changes made the improvement of the manuscript.

The research area and results are from the context of the manuscript can better understand.

However, authors are asked to carefully review the manuscript template and visual of the manuscript.
Figure 1 is on two pages.
Figure 3 - enlarge the text (captions) in the figure.

The results of the research and information value of the manuscript can be evaluated overall very well.

The manuscript can be published in the journal.

Author Response

Point 1: Figure 1 is on two pages.

Response 1: Thank you for your comments. I have already placed Figure 1 on the same page
Point 2: Figure 3 - enlarge the text (captions) in the figure.

Response 2: I have enlarged the text (captions) in the Figure 3.

Point 3: The results of the research and information value of the manuscript can be evaluated overall very well.

The manuscript can be published in the journal.

Response 3: Thank you for your comments.

Reviewer 3 Report

the authors carried out all the comments highlighted by the reviewers, thus the manuscript cam be accpted for publication, although again the manuscript is poor in terms of novelty/originality

Author Response

Point 1: the authors carried out all the comments highlighted by the reviewers, thus the manuscript cam be accepted for publication, although again the manuscript is poor in terms of novelty/originality

Response 1: Thank you for your comments.